# Cadmium Sorption on Alumina Nanoparticles and Mixtures of Alumina and Smectite: An Experimental and Modelling Study

Natalia Mayordomo [1,*], Tiziana Missana [2] and Ursula Alonso [2,*]

1   Helmholtz-Zentrum Dresden-Rossendorf e.V. (HZDR), Institute of Resource Ecology, Bautzner
    Landstrasse 400, 01328 Dresden, Germany
2   Centro de Investigaciones Energéticas, Mediaoambientales y Tecnológicas (Ciemat), Unidad de Fisicoquímica
    de Actínidos, Productos de Fisión y Migración de Radionucleidos, Avda Complutense 40,
    28040 Madrid, Spain; tiziana.missana@ciemat.es
*   Correspondence: n.mayordomo-herranz@hzdr.de (N.M.); ursula.alonso@ciemat.es (U.A.);
    Tel.: +49-351-260-2076 (N.M.); +34-91-3466139 (U.A.)

**Abstract:** Cadmium (Cd) is one of the most toxic transition metals for living organisms. Thus, effective measures to remediate Cd from water and soils need to be developed. Cd immobilization by alumina and mixtures of alumina and smectite have been analyzed experimentally and theoretically by sorption experiments and sorption modelling, respectively. Removal of aqueous Cd was dependent on pH and Cd concentration, being maximal for pH > 7.5. A two-site non-electrostatic sorption model for Cd sorption on alumina was developed and it successfully reproduced the experimental Cd immobilization on alumina. Cd sorption on mixtures of alumina and smectite were depending on pH, ionic strength, and alumina content in the mixture. Cd removal in mixtures increased with alumina content at high pH and ionic strength values. However, Cd sorption decreased with increasing alumina content under acidic conditions and low ionic strength. This effect was the result of alumina dissolution and the release of $Al^{3+}$ into the suspension at low pH values. Modelling of Cd sorption on mixtures of alumina and smectite was performed by considering the individual Cd sorption models for alumina and smectite. It could be shown that the contributions of the individual sorption models were additive in the model for the mixtures when the competition of $Al^{3+}$ with $Cd^{2+}$ for cation exchange sites in smectite was included.

**Keywords:** Cd; sorption model; retention; immobilization; remediation; heavy metals; $Al_2O_3$

## 1. Introduction

The International Agency for Research on Cancer (part of the World Health Organisation) classifies cadmium (Cd) in group 1 of species that are carcinogenic for humans [1], being related to bone, liver, muscle or lung diseases [2,3]. The main Cd intake source for humans is food. It is estimated that the daily dose that is ingested by humans is in the range of 10–35 µg, and the lethal oral dose is 350–3500 mg [4].

Therefore, Cd remediation strategies need to be developed to avoid the entrance and/or the migration of Cd in the biogeosphere. In fact, Cd remediation is a topic of great interest and numerous remediation strategies have been considered up to date, such as precipitation, coagulation, sorption, or separation [5,6].

One of the cheapest and most environmentally friendly options to remediate Cd-polluted areas is the use of naturally occurring sorbents, such as minerals [7–14]. In the last decades, an increasing number of publications have used nanoparticulate counterparts of minerals due to their good sorption capabilities [15,16].

Various review articles scrutinized the sorption capacities of different materials for Cd and their respective fits by using Langmuir and Freundlich isotherms [15,17]. Others are oriented towards summarizing the mechanistic interaction of Cd with various naturally occurring minerals [18], describing the formation of both inner- and outer-sphere complexes of Cd.

However, a small number of works report a combined experimental and theoretical mechanistic description of Cd sorption; such data are necessary to assess Cd remediation options in polluted areas. In order to study the sorption of Cd under more realistic conditions, it is crucial to elucidate the effect of other aqueous ions and the presence of multiple sorbents. For example, the role of chloride in Cd retention is not yet clear. Several Cd-Cl species can form in solution [19], but they do not exhibit a generalized behavior. In some cases, $Cl^-$ induces a higher Cd removal [20–23], while it seems to decrease Cd retention in other studies [24–27]. A recent publication states the importance of chloride and sulfate on the migration of Cd in soils and its subsequent uptake by plants [28].

In addition, in the environment, several minerals and living organisms coexist, and their presence might alter Cd sorption. Up to date, few works have studied the immobilization of Cd on mixtures of several sorbents. Some of them include the interaction of Cd with the following: clay and microorganisms [29], biochar and MgO composites [30], biochar and Fe/Mg composites, pyrogenic carbon and ferrihydrite [31], mixtures of components in reactive barriers [32], or mixtures of alumina and silica [33]. The development of sorption models to describe Cd sorption in mixtures of sorbents is essential to better understand Cd behavior in the environment.

This work aims to analyze Cd immobilization in $\gamma$-$Al_2O_3$ nanoparticles and in mixtures of FEBEX smectite and $\gamma$-$Al_2O_3$ nanoparticles with a combined experimental and theoretical approach. Both sorbents were selected because of their ubiquity and their good cation sorption capabilities ($\gamma$-$Al_2O_3$ nanoparticles [34,35] and FEBEX smectite [14,36–39]). Cd sorption on $\gamma$-$Al_2O_3$ nanoparticles, and in mixtures of smectite and $\gamma$-$Al_2O_3$ nanoparticles was studied in a wide range of pH values (3.0–11.0), Cd concentrations (from $1 \cdot 10^{-10}$ to $1 \cdot 10^{-3}$ M), $Cl^-$ concentrations (0, $1 \cdot 10^{-8}$, and $1 \cdot 10^{-3}$ M), ionic strengths (from $5 \cdot 10^{-4}$ to $1 \cdot 10^{-1}$ M), and mixture composition (from 0% $\gamma$-$Al_2O_3$ nanoparticles to 100% $\gamma$-$Al_2O_3$ nanoparticles). A model describing Cd sorption on $\gamma$-$Al_2O_3$ nanoparticles was developed. Cd sorption on mixtures of smectite and $\gamma$-$Al_2O_3$ nanoparticles was modelled by an additive approach using the sorption models of the individual components and by considering the chemistry of the aqueous phase.

## 2. Materials and Methods

### 2.1. General Experimental Conditions and Materials Used

All the experiments were conducted under equilibrium conditions with the atmosphere, and at ambient conditions in laboratories where the work with radioactive material is regulated.

Solutions were prepared in Milli-Q water (resistivity 18.2 MΩ·cm, Merck, Darmstadt, Germany).

The pH of solutions and suspensions was measured after daily calibration of the electrode and pH adjustment was carried out by adding aliquots of NaOH, HCl, or $HClO_4$ from $1 \cdot 10^{-2}$ M to 1 M to a buffer-containing suspension (more details about the buffers used are reported elsewhere [40]). Carbonate concentration in solution was measured in the suspensions at different pH, by ion chromatography (Dionex ICS-2000, Sunnyvale, USA) and the average value in solution was 1 mM $HCO_3^-$.

#### 2.1.1. Sorbents

Gamma alumina nanoparticles ($\gamma$-$Al_2O_3$ NPs, purchased in Sigma-Aldrich, St. Louis, MO, USA) were previously characterized by their isoelectric point ($pH_{IEP}$ 8.5), point of zero charge ($pH_{ZPC}$ 8.3), and specific surface area (136 $m^2$/g, calculated from nitrogen isotherms at 77 K) [41].

Smectite used in the experiments was obtained from FEBEX bentonite, a clay extracted from El Cortijo de Archidona (Almería, Spain), which is composed of 93% smectite and the following accessory minerals: 3% plagioclase (mixture of $NaAlSi_3O_8$ and $CaAl_2Si_2O_8$), 2% quartz ($SiO_2$), 1% cristobalite ($SiO_2$), and traces of calcite ($CaCO_3$), potassium feldspar ($KAlSi_3O_8$), and tridymite ($SiO_2$) [42]. FEBEX bentonite has been widely investigated with

the aim of understanding its behavior as a buffer material for a deep geological repository in crystalline rock [43]. For the experiments, Na-homoionized smectite was used following the procedure described elsewhere [34]. Na-homoionized smectite has been previously characterized with a cation exchange capacity (CEC) of $100 \pm 4$ meq/100 g and specific surface area (59 $m^2$/g, calculated from nitrogen isotherms at 77 K) [38].

For the sake of simplification, $\gamma$-$Al_2O_3$ NPs and Na-homoionized FEBEX bentonite will be hereafter referred as to *A* and *S*, respectively.

Suspensions of *A*, *S*, and *A/S* mixtures at a given $NaClO_4$ ionic strength were prepared as described elsewhere [34].

### 2.1.2. Adsorbate

Cd(II) was used as adsorbate in the experiments. Cd(II) was added as $CdCl_2$ or $Cd(ClO_4)_2$.

For the sorption experiments aliquots of a $^{109}CdCl_2$ stock solution (Isotope Products. Berlin, Germany) containing Cd-carrier were added. The stock solution had a total $1 \cdot 10^{-5}$ M Cd(II) concentration and it was dissolved in $1 \cdot 10^{-1}$ M HCl. Higher Cd concentrations were reached by adding aliquots of $CdCl_2$ (Sigma-Aldrich, St. Louis, USA) or $Cd(ClO_4)_2$ (Sigma-Aldrich, St. Louis, USA) stock solutions (ranging from $1 \cdot 10^{-6}$ M to $1 \cdot 10^{-1}$ M) dissolved in $1 \cdot 10^{-1}$ M HCl or $HClO_4$.

$^{109}Cd$ has a half-life of 426.6 days and it decays by electron capture, being detectable by gamma spectrometry between 60 and 463 keV.

### 2.2. Cadmium Sorption Experiments

Cd sorption experiments were prepared in 12 mL polypropylene centrifuge tubes, with a total suspension volume of 10 mL. The effect of several variables (ionic strength, chloride concentration, Cd concentration, and sorbent) on Cd removal was analyzed. In general, Cd sorption experiments were prepared by the following procedure. A sorbent suspension was added (0.5 g/L final solid to liquid ratio at given $NaClO_4$ concentration) to the centrifuge tube, followed by the addition of aliquots of buffers ($2 \cdot 10^{-3}$ M final concentration, as described in [40]) and Cd(II) stock solutions; then, the pH was adjusted if needed. After the required contact time of circular shaking, the Cd-containing suspensions were centrifuged at $21,500 \times g$ for 60 min. The supernatant was analyzed for Cd concentration (see Section 2.2.1) and pH.

Tables 1–3 summarize the experimental details of sorption experiments for pure sorbents (Tables 1 and 2) and *A/S* mixtures (Table 3).

**Table 1.** Experimental conditions of Cd sorption experiments by pure $\gamma$-$Al_2O_3$ NPs (*A*). Variable parameters are highlighted in bold. Chloride concentration is given by the Cd-stock solution addition. Parameters shown in bold font highlight the variables modified in the experiments.

| Experiment | Kinetics | pH Effect | Isotherm |
|---|---|---|---|
| Solid to liquid ratio of *A* | 0.5 g/L | 0.5 g/L | 0.5 g/L |
| $[Cd^{2+}]_0$ | $2 \cdot 10^{-9}$ M | $4.6 \cdot 10^{-8}$ M, $1 \cdot 10^{-5}$ M | **$1 \cdot 10^{-10}$ M–$1 \cdot 10^{-3}$ M** |
| pH | 5.5 | **3.0–11.0** | 9.8, 8.9, 6.1/7.2 |
| $[Cl^-]$ | $4 \cdot 10^{-9}$ M | 0 M, $10^{-8}$ M, $10^{-3}$ M | $1 \cdot 10^{-3}$ M |
| Contact time | **30 min–64 days** | 7 days | 7 days |
| Ionic strength $NaClO_4$ | $1 \cdot 10^{-1}$ M | $5 \cdot 10^{-4}$ M–$10^{-1}$ M | $1 \cdot 10^{-1}$ M/$1 \cdot 10^{-3}$ M |

**Table 2.** Experimental conditions of Cd sorption experiments by pure Na-homoionized smectite (*S*). Variable parameters are highlighted in bold. Contact time for the sorption experiments was set to seven days. Chloride concentration is given by the Cd-stock solution addition. Parameters shown in bold font highlight the variables modified in the experiments.

| Experiment | Ionic Strength Effect |
|---|---|
| Solid to liquid ratio of *S* | 0.5 g/L |
| $[Cd^{2+}]_0$ | $4.6 \cdot 10^{-8}$ M |
| pH | 4.5 |
| $[Cl^-]$ | $1 \cdot 10^{-3}$ M |
| Ionic strength NaClO$_4$ | **$5 \cdot 10^{-4}$ M–$10^{-1}$ M** |

**Table 3.** Experimental conditions of Cd sorption experiments by mixtures of γ-Al$_2$O$_3$ NPs (*A*) and Na-homoionized smectite (*S*). Variable parameters are highlighted in bold. Contact time for the sorption experiments was set to seven days. Chloride concentration is given by the Cd-stock solution addition. Parameters shown in bold font highlight the variables modified in the experiments.

| Experiment | pH Dependency | | Mixture Composition |
|---|---|---|---|
| *A*/*S* mixtures (wt.%) | 80*A*/20*S* | 50*A*/50*S* | **(100*A* → 100*S*)** |
| Total solid to liquid ratio | 0.5 g/L | 0.5 g/L | 0.5 g/L |
| $[Cd^{2+}]_0$ | $4.8 \cdot 10^{-8}$ M | $4.8 \cdot 10^{-8}$ M | $4.8 \cdot 10^{-8}$ M |
| pH | **3.0–11.0** | **3.0–12.0** | 4.3/8.0 |
| Ionic strength NaClO$_4$ | $1 \cdot 10^{-1}$ M | **$5 \cdot 10^{-4}$ M–$1 \cdot 10^{-1}$ M** | $1 \cdot 10^{-2}$ M/$1 \cdot 10^{-1}$ M |
| $[Cl^-]$ | $1 \cdot 10^{-3}$ M | $1 \cdot 10^{-3}$ M | $1 \cdot 10^{-3}$ M |

It was verified that neither the presence of buffer-affected Cd sorption on the sorbents nor sorption of Cd on the centrifuge tubes occurred.

2.2.1. Cadmium Sorption Quantification

Three aliquots (2 mL) of the supernatant were extracted to determine the final [109]Cd activity by gamma counting with a NaI detector (Packard Autogamma COBRA2, PerkinElmer, Waltham, MA, USA).

It was considered that natural Cd and [109]Cd have the same affinity for the sorbent surfaces. The experimental error of the gamma measurements never exceeded 2% of the measured value.

Experiments to evaluate the effect of chloride on Cd removal were performed with Cd(ClO$_4$)$_2$ (Sigma-Aldrich, St. Louis, USA) instead of CdCl$_2$; the rest of the conditions were the same as described in Table 1. Aliquots of supernatant were extracted and acidified with HNO$_3$ (Sigma-Aldrich, St. Louis, USA) prior their quantification for Cd in solution by inductively coupled plasma mass spectrometry (Thermo Fischer Scientific X Series II, Waltham, USA).

Cd removal was calculated as percentage of Cd removed (%$Cd_{sorbed}$) and distribution coefficient ($K_D$ in mL/g) using Equations (1) and (2), respectively.

$$\%Cd_{sorbed} = \frac{[Cd]_0 - [Cd]_{eq}}{[Cd]_0} \cdot 100 \tag{1}$$

$$K_D = \frac{[Cd]_0 - [Cd]_{eq}}{[Cd]_{eq}} \cdot \frac{V}{m} \tag{2}$$

where $[Cd]_0$ is the initial Cd concentration (in Bq/L or M), $[Cd]_{eq}$ is the final Cd equilibrium concentration in solution (in Bq/L or M), $V$ is the volume of the suspension (in mL), and $m$ is the mass of sorbent (in g).

### 2.3. Cadmium Sorption Modelling

The geochemical code CHESS 2.4 was used to perform the model calculations [44].

Table 4 summarizes the Cd thermodynamic data of Cd solid and aqueous speciation considered in the calculations. Figures S1–S3 show the Cd speciation as a function of pH for the most studied experimental conditions in this work ($[Cd^{2+}]_0 = 4.6 \cdot 10^{-8}$ M, $[NaClO_4] = 1 \cdot 10^{-1}$ M, $[HCO_3^-] = 1 \cdot 10^{-3}$ M, and $[NaCl] = 1 \cdot 10^{-3}$ M).

**Table 4.** Cd species considered in modelling. Thermodynamic data were taken from the IUPAC Cd database [45], * from [46], and ** from [14]. Uncertainties of the thermodynamic data can be found in the respective references. Species with similar chemical composition are defined as aqueous (aq) or solid (s) to avoid misunderstanding. Words in bold font highlight the aqueous and solid species used in the model.

| Species | Code Definition | $\log_{10} K^{\circ}$ |
|---|---|---|
| **Aqueous** | | |
| $Cd^{2+}$ | Basis species | |
| $CdCl^+$ | 1 Cd[2+], 1 Cl[−] | 1.98 |
| $CdCl_2$ (aq) | 1 Cd[2+], 2 Cl[−] | 2.64 |
| $CdCl_3^-$ | 1 Cd[2+], 3 Cl[−] | 2.30 |
| CdCl(OH) (aq) | 1 Cd[2+], 1 Cl[−], 1 $H_2O$, −1 H[+] | −7.4328 * |
| $CdHCO_3^+$ | 1 Cd[2+], 1 $HCO_3^-$ | 1.50 * |
| $CdCO_3$ (aq) | 1 Cd[2+], 1 $HCO_3^-$, −1 H[+] | −5.9288 |
| $Cd(CO_3)_2^{2-}$ | 1 Cd[2+], 2 $HCO_3^-$, −2 H[+] | −14.4576 |
| $CdOH^+$ | 1 Cd[2+], 1 $H_2O$, −1 H[+] | −9.91 |
| $Cd(OH)_2$ (aq) | 1 Cd[2+], 2 $H_2O$, −2 H[+] | −20.19 |
| $Cd(OH)_3^-$ | 1 Cd[2+], 3 $H_2O$, −3 H[+] | −33.50 |
| $Cd(OH)_4^{2-}$ | 1 Cd[2+], 4 $H_2O$, −4 H[+] | −47.28 |
| $Cd_2OH^{3+}$ | 2 Cd[2+], 1 $H_2O$, −1 H[+] | −8.73 |
| $Cd_4(OH)_4^{4+}$ | 4 Cd[2+], 4 $H_2O$, −4 H[+] | −31.8 |
| **Solid** | | |
| Cd (s) | 1 Cd[2+], 1 $H_2O$, −2 H[+], −0.5 $O_2$(aq) | −56.6 |
| $CdCl_2$ (s) | 1 Cd[2+], 2 Cl[−] | 0.674 * |
| $CdCl_2{:}H_2O$ (s) | 1 Cd[2+], 2 Cl[−], 1 $H_2O$ | 1.6747 * |
| CdCl(OH) (s) | 1 Cd[2+], 1 Cl[−], 1 $H_2O$, −1 H[+] | −3.543 |
| Otavite ($CdCO_3$ (s)) | 1 Cd[2+], 1 $HCO_3^-$, −1 H[+] | −0.1 ** |
| $Cd(OH)_2$ (s) | 1 Cd[2+], 2 $H_2O$, −2 H[+] | −13.72 * |
| Monteponite (CdO (s)) | 1 Cd[2+], 1 $H_2O$, −2 H[+] | −15.097 |

A non-electrostatic sorption model (SM) for Cd sorption on *S* has been recently described [14]. For the sake of simplicity, non-electrostatic sorption models (SMs) were also developed to predict Cd sorption on pure *A* and *A*/*S* mixtures (Table 5).

SMs of Cd on *A*/*S* mixtures were described by considering an additive approach, i.e., the sum of the individual Cd sorption on *A* and *S*. The SM of Cd on *A*/*S* mixtures took into account the water chemistry. In brief, the water chemistry of the mixture considers the initial concentration of Cd in solution (in M), the concentration of the sorbents (in g/L), the concentration of $NaClO_4$ (in M), the concentration of $Cl^-$ (in M), and other secondary processes that influence the chemistry in solution as described in [34]: the dissolution of *A*, the $Al^{3+}$ cation exchange in *S*, and carbonate concentration ($[HCO_3^-] = 1 \cdot 10^{-3}$ M).

The SMs for Cd sorption on the pure sorbents were described by taking into account the sorption properties of the solids. *A* and *S* contain amphoteric surface sites that (de)protonate as a function of pH and their charge changes accordingly [47]. *A* and *S* present both strong and weak surface sites to sorb anionic and cationic species. *S* does not only contain strong and weak surface sites but it also has cationic exchange sites due to its structural properties [48], which enables cation exchange reactions of pollutants with *S*.

Amphoteric surface sites can be protonated and deprotonated depending on the pH value. The protonation and deprotonation reactions are:

$$XOH + H^+ \rightleftarrows XOH_2^+ \tag{3}$$

$$XOH \rightleftarrows XO^- + H^+ \tag{4}$$

where $X$ represents the surface active sites. Their equilibrium reactions are defined as:

$$K_+ = \frac{[XOH_2^+]}{[XOH][H^+]} \tag{5}$$

$$K_- = \frac{[XO^-][H^+]}{[XOH]} \tag{6}$$

where $K_+$ and $K_-$ are the protonation and deprotonation equilibrium constants, respectively. The protonation and deprotonation constants as well as the concentration of the surface active sites for $A$ and $S$ are summarized in Table 5.

The interaction of Cd with active surface sites can take place through different mechanisms. This includes inner-sphere, outer-sphere, and ternary surface complexation. In general, Cd sorption on surface sites can be described by the following reaction:

$$XOH + Cd^{2+} \rightleftarrows XOCd^+ + H^+ \tag{7}$$

For the development of the SM of Cd on $A$ in this work, we have assumed Cd sorption on both strong and weak sites of the $A$ surface and the reactions considered in the theoretical description of Cd sorption are summarized in the section "Cd sorption by surface complexation" of Table 5.

In addition, the sorption of the main anionic species ($Cl^-$, $ClO_4^-$, and $HCO_3^-$) on alumina has been considered in the models. They are expressed by:

$$XOH_2^+ + Cl^- \rightleftarrows XOH_2Cl \tag{8}$$

$$XOH_2^+ + ClO_4^- \rightleftarrows XOH_2ClO_4 \tag{9}$$

$$XOH_2^+ + HCO_3^- \rightleftarrows XOH_2HCO_3 \tag{10}$$

Such anionic complexation reactions on alumina were previously reported [41,49] and they are summarized in the section "anion sorption on $A$" of Table 5.

The cation exchange sites available in Na-homoionized smectite allows for the exchange of Na by other cations present in solution as follows:

$$Na \equiv Y + B^{b+} \rightleftharpoons B \equiv Y_b + bNa^+ \tag{11}$$

where $\equiv Y$ represents the cation exchange site, and $B$ is a cation of $b$ valence.

The selectivity coefficient ($_{Na}^{B}K_c$) representative of the cation exchange of Na by $B$ is defined, according to Gaines-Thomas, as:

$$_{Na}^{B}K_c = \frac{(N_B)[\gamma_{Na}]^b}{(N_{Na})^b[\gamma_B]} \tag{12}$$

where $\gamma_{Na}$ and $\gamma_B$ represent the activity coefficients of Na and *B*, respectively, and $N_{Na}$ and $N_B$ are their fractional occupancies on the solid. The fractional occupancy of a general element *i* ($N_i$) can be calculated as:

$$N_i = \frac{eq_{i,\,sorbed}/m}{CEC} \tag{13}$$

where $eq_{i,\,sorbed}/m$ is the equivalent of *i* sorbed per mass (*m*) of sorbent (in eq/g) and *CEC* is the cation exchange capacity of the solid (in eq/g).

One way to determine $^B_{Na}K_c$ experimentally was suggested by Bradbury and Baeyens [50]. Applied to the Na-homoionized smectite selected for this work, when the concentration $Na^+$ is much higher than the concentration of cation $B^{b+}$ in Equation (12), $^B_{Na}K_c$ can be calculated using the experimental distribution coefficient ($K_D$, Equation (2)):

$$^B_{Na}K_C = \frac{(K_D b)}{CEC} \frac{\{\gamma_{Na}\}^b}{\{\gamma_b\}}[A]^b \tag{14}$$

where the activity coefficients of Na and B ($\gamma_{Na}$ and $\gamma_B$) are calculated by using the Davies approximation for ionic strengths (*I*) below 0.5 M in aqueous systems at room temperature. The activity coefficient of an ion *i* can be calculated using the following equation.

$$-log\gamma_i = 0.5085z_i^2\left[\frac{\sqrt{I}}{1+\sqrt{I}} - 0.3I\right] \tag{15}$$

where $z_i$ is the charge of the ion *i*.

However, $^B_{Na}K_c$ cannot be direcly used in geochemical codes. The value entered in the geochemical code to account for the cation exchange, the exchange code, $K_{EX}$, can be calculated for the Na-homoionized smectite as follows:

$$^B_{Na}K_{EX} = {}^B_{Na}K_C \cdot (CEC)^{1-b} \cdot \frac{1}{b} \cdot (s/l)^{1-b} \tag{16}$$

where *s/l* is the concentration of the solid in suspension in (g/L). For further understanding of this equation, please refer to [50].

$^B_{Na}K_c$ values, when *B* is $Cd^{2+}$ or $Al^{3+}$, were defined in [34,49] and are summarized in the section "cation exchange in *S*" of Table 5. The values of $K_{EX}$ for the cation exchange of $Cd^{2+}$ and $Al^{3+}$ of *S*, which was used to describe the Cd sorption models in *S* and *A/S* mixtures will be specified below.

**Table 5.** Parameters and reactions used to describe Cd sorption on γ-Al$_2$O$_3$ NPs (*A*) and Na-homoionized FEBEX bentonite (*S*) in the CHESS V.2 software. *CEC* is the cation exchange capacity, *X* represents the surface active sites of γ-Al$_2$O$_3$ NPs (*A*) or Na-homoionized FEBEX smectite (*S*), ≡*Y* represents the cation exchange site of *S*, *s* and *w* stands for strong and weak surface sites, respectively. (*) Data obtained in this study. (§) Selectivity coefficient (log$_{10}$ *Kc*) related to the Na$^+$ exchange in the clay by Cd$^{2+}$ or by Al$^{3+}$, the value entered in the CHESS V.2 software was calculated using equation 16. The log$_{10}$ K° values were derived by sorption modelling and are not the result of experiments, thus, no uncertainties are provided.

| Parameter | *A* | *S* |
|---|---|---|
| CEC (meq/100 g) | | 100 ± 4 [38] |
| $[X^S OH]$ (µeq/m$^2$) | 9.5·10$^{-3}$ [51] | 3.4·10$^{-2}$ [37] |
| $[X^W OH]$ (µeq/m$^2$) | 1.1 [41] | 1.02 [52] |
| BET (m$^2$/g) | 136 [41] | 59 [38] |

**Table 5.** *Cont.*

| Species and reactions | $A$ $\log_{10} K°$ | $S$ $\log_{10} K°$ |
|---|---|---|
| **Surface sites acidity constants** | | |
| $X^sOH + H^+ \rightleftharpoons X^sOH_2^+$ | 6.90 [51] | 4.8 [37] |
| $X^sOH - H^+ \rightleftharpoons X^sO^-$ | −9.7 [51] | −9.9 [37] |
| $X^wOH + H^+ \rightleftharpoons X^wOH_2^+$ | 6.90 [41] | 5.3 [52] |
| $X^wOH - H^+ \rightleftharpoons X^wO^-$ | −9.7 [41] | −8.4 [52] |
| **Cation exchange in $S$** | | |
| $2Na \equiv Y + Cd^{2+} - 2Na^+ \rightleftharpoons Cd \equiv (Y)_2$ | - | 0.8 § [14] |
| $3Na \equiv Y + Al^{3+} - 3Na^+ \rightleftharpoons Al \equiv (Y)_3$ | - | 1.5 § [34] |
| **Cd sorption by surface complexation** | | |
| $X^sOH - H^+ + Cd^{2+} \rightleftharpoons X^sO - Cd^+$ | 0.1 * | −1.4 [14] |
| $X^wOH - H^+ + Cd^{2+} \rightleftharpoons X^wO - Cd^+$ | −2.25 * | −2.51 [14] |
| $X^sOH + Cd^{2+} \rightleftharpoons X^sOH - Cd^{2+}$ | 7.9 * | 6.10 [14] |
| $X^wOH + Cd^{2+} \rightleftharpoons X^wOH - Cd^{2+}$ | 5.4 * | 4.14 [14] |
| $X^sOH + Cd^{2+} + H_2O - 2H^+ \rightleftharpoons X^sO - CdOH$ | −12.5 * | −11.66 [14] |
| $X^wOH + Cd^{2+} + H_2O - 2H^+ \rightleftharpoons X^wO - CdOH$ | −13.2 * | −11.88 [14] |
| Empirical Cd sorption at low pH | $\log_{10}$ Kd (mL/g) = 2.2 * | - |
| **Anion sorption on $A$** | | |
| $X^{s,w}OH + H^+ + HCO_3^- \Leftrightarrow X^{s,w}OH_2HCO_3$ | 11.50 [41] | - |
| $X^{s,w}OH + H^+ + Cl^- \Leftrightarrow X^{s,w}OH_2Cl$ | 9.2 [49] | - |
| $X^{s,w}OH + H^+ + ClO_4^- \Leftrightarrow X^{s,w}OH_2ClO_4$ | 8.5 [41] | - |

## 3. Results and Discussion

### 3.1. Cadmium Sorption on Alumina Nanoparticles: Experiments and Modelling

3.1.1. Effect of Time

The kinetics of Cd sorption on alumina are shown in Figure 1.

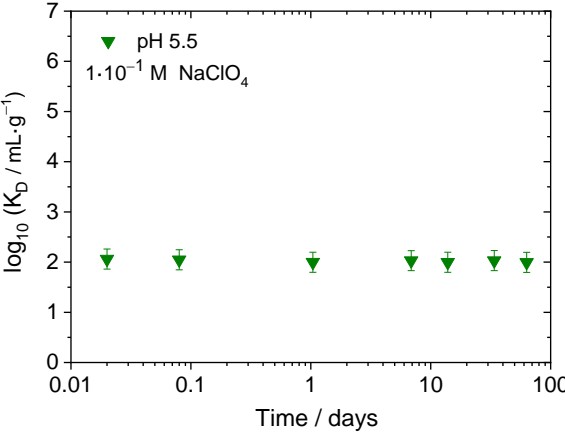

**Figure 1.** Cadmium sorption (as distribution coefficient $K_D$) on alumina nanoparticles ($A$) as a function of time. [$A$] = 0.5 g/L, ionic strength $1·10^{-1}$ M NaClO$_4$, pH 5.5, and [Cd]$_0$ = $4·10^{-9}$ M.

Figure 1 shows that sorption of Cd on $A$ is constant ($\log_{10} K_D \approx 2.2$) regardless of contact time. The sorption equilibrium is reached after 30 min of interaction. This is in agreement with the fast Cd sorption on alumina observed by Sen et al. [53], occurring in the first five minutes of Cd-alumina interaction. The following Cd sorption experiments were in contact for seven days with the aim of comparing the Cd sorption behavior on $A$ with that reported for $S$ [14].

3.1.2. Effect of pH, Ionic Strength, Chloride Concentration, and Cadmium Concentration

Various factors on the Cd removal by *A* have been studied. The effect of pH and ionic strength on Cd sorption is shown in Figures 2 and 3.

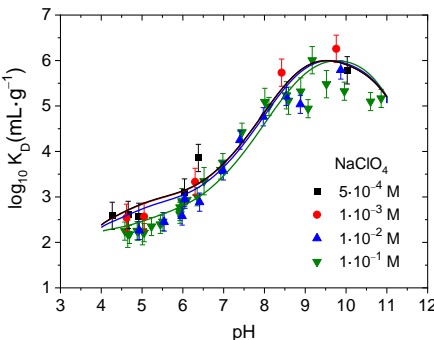

**Figure 2.** Cadmium sorption (as distribution coefficient $K_D$) on alumina nanoparticles (*A*) as a function of pH evaluated at different $NaClO_4$ concentrations (details in figure legend). Colored solid lines represent the Cd sorption model calculated using the parameters shown in Table 5 for different ionic strengths according to the figure legend. [*A*] = 0.5 g/L, contact time = 7 days, and $[Cd]_0 = 4.6 \cdot 10^{-8}$ M.

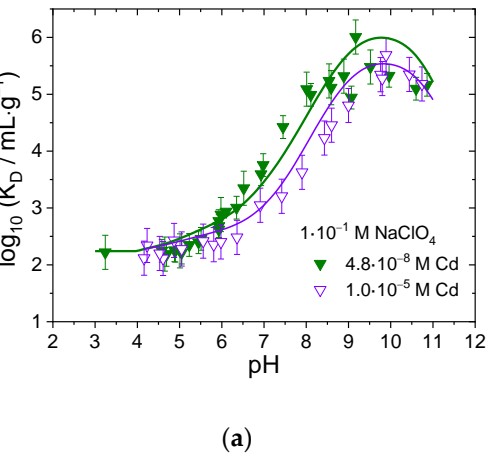

(**a**)

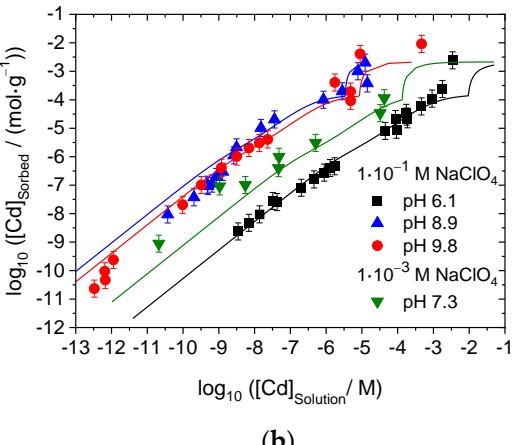

(**b**)

**Figure 3.** (**a**) Cadmium sorption (as distribution coefficient $K_D$) on alumina nanoparticles (*A*) as a function of pH values at $1 \cdot 10^{-1}$ M $NaClO_4$ and at different Cd concentrations (detailed in figure legend). (**b**) Cadmium sorption (as mol of Cd per gram of *A*) on alumina nanoparticles (*A*) as a function Cd concentration at different $NaClO_4$ ionic strengths and pH values (detailed in figure legend). Colored solid lines represent the model of Cd sorption that was calculated using the parameters shown in Table 5 for given conditions according to the figure legend. [*A*] = 0.5 g/L, contact time = 7 days, and $[Cd]_0$ from $1 \cdot 10^{-10}$ M to $1 \cdot 10^{-3}$M.

Cd sorption on alumina increases with increasing pH, being higher than 90% $Cd_{sorbed}$ at pH > 7.4. The influence of ionic strength on Cd removal by *A* is minimal, and it can only be observed at low pH values (pH < 7), when Cd sorption represents $\log_{10} K_D = 2.2$ (≈10%$Cd_{sorbed}$) at $1 \cdot 10^{-1}$ M and $1 \cdot 10^{-2}$ M ionic strengths, whereas at ionic strengths $1 \cdot 10^{-3}$ M and $5 \cdot 10^{-4}$ M the value of $K_D$ is slightly higher ($\log_{10} K_D = 2.6$, which represents ≈20% $Cd_{sorbed}$). At pH > 8.0, the variability of Cd sorption data can be simply due to experimental error since the values of % $Cd_{sorbed}$ range from 95% to 100%. The effect of ionic strength on pollutant sorption gives hints about the pollutant surface complexation mechanisms [54]. A decrease in pollutant sorption with increasing ionic strength is related to outer-sphere complexation, whereas inner-sphere complexation occurs when ionic strength changes do not alter pollutant sorption. Thus, Cd sorption on alumina at pH < 7 Cd

sorption on alumina could be partially represented by outer-sphere complexation, whereas at pH > 7 Cd sorption on alumina occurs by inner-sphere complexation.

The sorption of Cd on *A* follows the general sorption trend of cations on metal oxides [55]. However, removal of 10 to 20% of aqueous Cd was observed under acidic conditions. Cd sorption on $\gamma$-$Al_2O_3$ at acidic conditions was also reported in [53,56], but this was not observed by others at pH < 5 [57]. We have observed such unusual cation sorption at low pH for $Sr^{2+}$ and $Fe^{2+}$ when using alumina nanoparticles as sorbent [34,35]. The overall zeta potential of *A* under those pH values is positive [41,58], and the interaction between $Cd^{2+}$ and *A* should be electrostatically hindered, but some surface sites of *A* can be negatively charged. Nanoparticles possess a higher area-to-volume ratio and specific surface area than their bulk counterparts. A hypothesis to explain such unusual cation sorption at low pH values is that these nanoparticle properties favor the interaction of $Cd^{2+}$ with *A* despite its overall positive charge.

The influence of Cd concentration on its immobilization on *A* was studied at different ionic strengths and pH values and the results are depicted in Figure 3.

The Cd sorption curves slightly differ with the increase of the initial Cd concentration in solution (Figure 3a). This effect is especially remarkable at pH values between pH 6 and pH 8, when $\log_{10} K_D$ is 0.5 units lower at higher Cd concentrations than at lower Cd concentrations (Figure 3a). Cd sorption isotherms show an increase in Cd immobilization with increasing Cd concentration (Figure 3b), but this trend is not linear, and the slope resulting from the fit of $\log_{10} [Cd]_{Sorbed}$ vs $\log_{10} [Cd]_{Solution}$ leads to a value lower than one. These observations indicate that Cd sorption on *A* is dependent on Cd concentration and suggest that either more than one surface site of *A* is involved in Cd sorption or several different Cd species are sorbed on *A*. Figure S7 represents the individual contribution of the Cd complexes to the isotherm curves.

Other factors that influence Cd sorption on *A* are the formation of Cd-containing precipitates, the presence of Cd-Cl species, and the aging of *A*.

Some authors have observed the formation of a new solid phase, layered double hydroxides (LDH), when divalent metals interact with trivalent metal oxides under neutral to alkaline pH conditions [59,60]. Such a solid formation is visible to the naked eye when it occurs [35]. Some authors discard the formation of Cd-Metal(III) LDH [61,62], whereas another publication confirms its formation [63].

The formation of Cd-Al LDH was observed at high Cd concentration (in the mM range) [63]. In the present work, we have generally worked with low Cd concentration (below $\mu$M range). The formation of Cd-Al LDH would increase Cd sorption due to precipitation with increasing Cd concentration. However, in this work we observed the opposite behavior, as depicted in Figure 3a. Therefore, the formation of Cd(II)-Al(III) layered double hydroxide was excluded and was not considered in the SM.

Several papers report that Cd-chloride species can influence Cd sorption and are either increasing [20–23,64,65] or decreasing it [24–27,66,67]. Furthermore, alumina transforms to gibbsite ($Al(OH)_3$) over time [68], which might modify the Cd sorption conditions. Therefore, additional experiments were carried out with the aim of evaluating whether the presence of chloride or the age of *A* suspensions influenced Cd removal (results are plotted in Figures S4 and S5).

However, it was observed that neither the concentration of chloride nor the aging time of the *A* suspensions had an effect on Cd sorption on *A*. This confirms that Cd-chloride species are not involved in Cd sorption and that the possible alumina phase change does not influence Cd sorption.

### 3.1.3. Cd Sorption Model on Alumina

A non-electrostatic SM for Cd sorption on *A* has been developed by considering two sorption sites —weak and strong (*w* and *s*)— with different affinities for Cd, according to the results obtained in Figure 3. The sorption of Cd-chloride species on *A* and the formation of Cd-Al LDH were not considered in the SM.

Two-site models have been previously described for metal oxides [69] and aluminium oxide [34,70–73]. None of the reported Cd sorption studies onto any kind of $Al_2O_3$ phase considered a two-site model, probably because reported Cd sorption studies on $Al_2O_3$ were carried out with higher initial Cd concentration ($>10^{-6}$ M) [53,56,57,64,68,74–80] than in the present work. Strong sorption sites prevail at low adsorbate concentration. Thus, the contribution of strong sorption sites might have been overlooked in previous works.

The SM was developed considering similar equilibria to those suggested in the SM of Cd sorption on *S* [14]: the sorption of $Cd^{2+}$ and $CdOH^+$ on negatively charged surface sites on *A* ($X^{s,w}O\text{-}Cd^+$ and $X^{s,w}O\text{-}CdOH$) and the sorption of $Cd^{2+}$ on neutral surface sites on *A* ($X^{s,w}OH\text{-}Cd^{2+}$). However, the model was unable to simulate the experimental Cd sorption for pH < 5.0. With the aim to reproduce theoretically the sorption of Cd under acidic conditions, an empirical Cd sorption value of $\log_{10} K_D \approx 2.2$ (Table 5) was considered. This approach was used in [34], as no experimental evidence was available to describe the nature of the unusual cation sorption on *A*. The SM could reproduce the experimental data obtained for Cd sorption on *A* (see solid lines in Figures 1–3 and Figures S4–S7). Figures S6 and S7 show the individual contributions of the equilibria, which were considered to describe the total Cd sorption on *A* in this SM.

Although the formation of a ternary complex of Cd to *A* surface—i.e., enabled by $ClO_4^-$, which is the main anion in solution—can be a feasible explanation for the sorption of Cd at low pH values; however, we do not have experimental evidence proving that this occurs.

### 3.2. Cadmium Sorption on Mixtures of Alumina Nanoparticles and Smectite: Experiments and Modelling

Cd sorption in *A*/*S* mixtures (in wt. %) was studied as a function of ionic strength, pH, and mixture composition, as summarized in Table 3. Sorption of Cd on 50*A*/50*S* mixtures at different ionic strengths is shown in Figure 4. Sorption of Cd on 50*A*/50*S* and 80*A*/20*S* at $1\cdot10^{-1}$ M ionic strength is shown in Figure S8. Cd removal by *A*/*S* mixtures as a function of *A* content at fixed pH (4.1 and 8.0) and ionic strength ($1\cdot10^{-2}$ M and $1\cdot10^{-1}$ M NaClO$_4$) is shown in Figure 5.

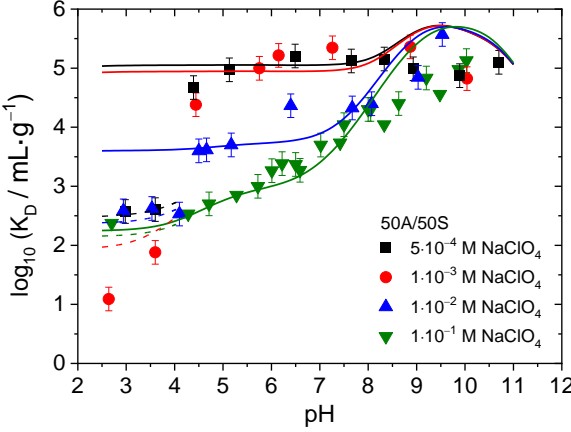

**Figure 4.** Cadmium sorption (as distribution coefficient $K_D$) on mixtures of alumina nanoparticles (*A*) and Na-homoionized smectite (*S*) as a function of pH for different ionic strengths (detailed in figure legend). Colored lines represent the Cd sorption on *A*/*S* mixtures calculated using the data collected in Table 5. Solid lines represent the sorption of Cd without including $Al^{3+}$ competition. Dashed lines represent the sorption of Cd including 2 mg/L of $Al^{3+}$. $[A/S]_{total}$ = 0.5 g/L, 50% *A* and 50% *S*, contact time = 7 days, and $[Cd]_0 = 4.8\cdot10^{-8}$ M.

Cadmium sorption on 50*A*/50*S* mixtures increases with increasing pH. At pH > 9.0 Cd sorption does not depend on ionic strength, whereas at pH < 9.0 Cd sorption increases with decreasing ionic strength (Figure 4). This effect is due to the contribution of cation exchange in *S*.

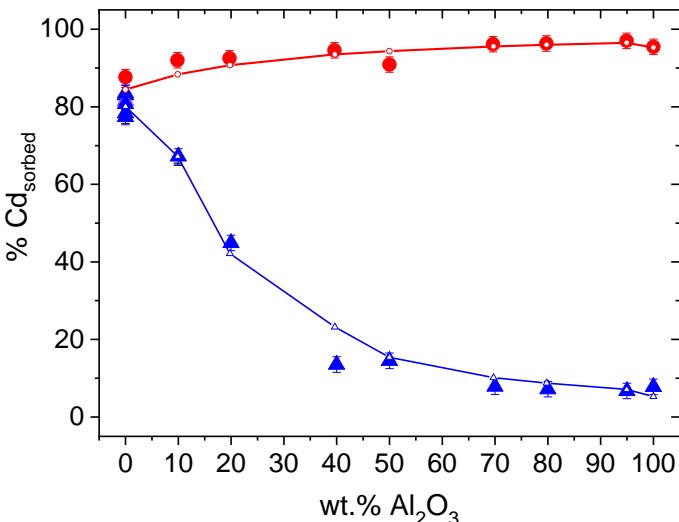

**Figure 5.** Cadmium sorption (as distribution coefficient $K_D$) on mixtures of alumina nanoparticles (*A*) and Na-homoionized smectite (*S*) as a function of A content in wt.% for (▲) $1.0 \cdot 10^{-2}$ M at pH 4.1, and (●) $1.0 \cdot 10^{-1}$ M at pH 8.0. Empty plots and colored lines represent the Cd sorption on *A/S* mixtures calculated using the data collected in Table 5 and including 1 mg/L of $Al^{3+}$ for 10% A mixture and 2 mg/L of $Al^{3+}$ for the rest of *A/S* mixtures. $[A/S]_{total} = 0.5$ g/L, *S*, contact time = 7 days, and $[Cd]_0 = 4.8 \cdot 10^{-8}$ M.

The first attempt to apply an additive SM to fit the data was successful for $I \geq 1 \cdot 10^{-2}$ M, but the model overestimated Cd sorption for $I \leq 1 \cdot 10^{-3}$ M. The factor in the SM for both *A* and *S*, which is the most affected by ionic strength is cation exchange in *S*. Therefore, Cd sorption on *S* as a function of ionic strength and pH 4.5 was to evaluate possible influences on Cd sorption. It was observed that when the selectivity coefficient recently reported for Cd in [14] ($\log_{10} K_c = 0.8$) was considered in the SM, Cd sorption was overestimated for $I < 5.0 \cdot 10^{-2}$ M. This is most likely due to the presence of other cations in solution—coming from *S* dissolution or from impurities—that compete with Cd for cation exchange sites in *S*. Therefore, the selectivity coefficient ($\log_{10} K_c$) was modified to an "apparent selectivity coefficient" that allowed for the simulation of Cd removal at low pH values. The model of Cd sorption, considering the equilibria constants in Table 5, is shown in solid lines in Figure 4. The model simulates the experimental Cd sorption data, except for pH < 4.0. Under those pH values, it is known that *A* dissolution occurs [34]. In the SM of Cd sorption on *A*, the dissolution of *A* was not considered. The formation of *A* is described by the following reaction.

$$2Al^{3+} + 3H_2O \rightarrow Al_2O_3 + 6H^+ \tag{17}$$

The most accepted equilibrium formation constant of *A* ($\log_{10} K_{Form} = -17.8$ [81]) clearly overestimates the dissolution of *A* at pH < 5.0 and pH > 11 [34]. The $\log_{10} K_{Form}$ value needed to simulate Cd sorption on *A/S* mixtures at pH < 4.0 in Figure 4 would be

$\log_{10} K_{Form} = -9.0$, whereas the reported *A* formation constants range from $\log_{10} K_{Form} = -16.1$ to $-19.1$ [81–83]. This was perhaps due to the kinetic effects of *A* dissolution. To simulate the Cd sorption on *A/S* mixtures, it was necessary to add a given aluminum concentration to the solution that competed with $Cd^{2+}$ for cation exchange in smectite (Table 5) in order to fit the theoretical data with the experimentally measured Cd sorption data. The added $Al^{3+}$ concentration in the model is described in the captions of Figures 4 and 5.

These observations imply that Cd sorption in *A/S* mixtures is very susceptible to the presence of competitive cations coming from mineral dissolution, either from *A* or *S*.

Cd sorption at $I = 1.0 \cdot 10^{-1}$ M is very similar for pure *A* and *S*, being higher in *S* under acidic and neutral pH values, and slightly higher in *A* under alkaline conditions (Figure S8). The Cd immobilization in *A/S* mixtures exhibits an intermediate behavior, showing an

improved Cd removal under (i) acidic conditions with increasing $S$ content, and (ii) under alkaline conditions with increasing $A$ content in the $A/S$ mixtures (Figure S8).

The effect of $A$ content in the mixtures on Cd removal was studied in detail at two pH values and two ionic strengths (Figure 5).

The composition of the $A/S$ mixture influences the retention of Cd. At pH 8 and $1 \cdot 10^{-1}$ M NaClO$_4$, Cd sorption on $S$ (%Cd$_{sorbe}$ = 87%) is enhanced by increasing the content of $A$ (%Cd$_{sorbe}$ = 97%). However, at acidic pH and $I = 1 \cdot 10^{-2}$ M, Cd sorption decreases from %Cd$_{sorbe}$ = 84% in 100$S$ to %Cd$_{sorbe}$ = 8% by increasing the amount of $A$ in the $A/S$ mixtures. The SM was additive and reproduced the experimental Cd sorption when the given Al$^{3+}$ content was considered in solution. According to the results in Figures 4 and 5, the use of $A/S$ mixtures to remediate Cd-polluted areas needs to be assessed depending on the chemical conditions of the environment. The addition of $S$ to alumina improves Cd sorption at low pH values (from pH 4.0 to pH 7.0) and low ionic strengths ($\leq 1 \cdot 10^{-3}$ M), while adding $A$ to $S$ enhances Cd removal at high pH > 8.5. Another factor to be considered in the removal of Cd by minerals is their size in suspension, since it has been known that colloids in suspension (particle size in suspension below 1 μm) can be responsible for the migration of pollutants [84,85]. We have observed that $A/S$ mixtures lead to an increase in the colloidal size at pH < 9.0 with increasing $A$ content and ionic strengths [51]. Therefore, the use of $A/S$ mixtures as Cd scavengers would not only improve Cd sorption but it would limit the migration of pollutants driven by colloids.

The literature dealing with Cd sorption and additive SMs by mixtures of two or more than two components present inconsistent results.

Some studies dealing with Cd retention on different mixed systems could explain Cd sorption results by additivity of the sorption on independent components: in iron oxide, silicon oxide and quartz mixture [86], in mixtures of hydrous ferric oxide, kaolinite and bacteria (*B. suctilis*) [87], and in mixtures of alumina and silica [33].

In other studies dealing with Cd sorption on mixtures of silicon and iron oxides showed that Cd retention was enhanced at low pH, which was not observed in the absence of silicon [88]. The improvement of Cd removal by the mixture was explained by the presence of silicate, which acted as a bridge to bind Cd to the surface. Other studies on Cd removal using mixtures of oxides (goethite and aluminium oxide) showed Cd sorption was not additive [89], and it was attributed to Al$^{3+}$ ions covering and modifying the iron oxide surface, changing its surface properties [90].

Thus, it is crucial to assess the presence of additional ions in solution (coming from mineral dissolution or from the water chemistry) in order to evaluate the performance of Cd sorption using mixtures since they might improve or worsen the removal of Cd.

## 4. Conclusions

Cd sorption on $A$ is fast and nearly independent of ionic strength, but it is dependent on pH (increasing with pH) and Cd concentration. Cd sorption is close to 100% at pH > 7.5. An increase in initial Cd concentration decreases the distribution coefficient ($\log_{10} K_D$) by 0.5 units at pH 5.0 to 8.0. This indicates that $A$ presents two surface sites to bind Cd. Our work is the first one reporting strong sorption sites in Cd removal when using metal oxides as Cd scavengers.

We have developed a two-site non-electrostatic SM, including the sorption of Cd$^{2+}$ on negative and neutral surface sites, the sorption of CdOH$^+$ on negative surface sites, and an empirical Cd sorption of $\log_{10} K_D = 2.2$ to describe the unusual Cd$^{2+}$ sorption under low pH values.

Cd sorption on $A/S$ mixtures is maximum for pH > 4.5 at low ionic strengths ($\leq 1 \cdot 10^{-3}$ M). Two different effects could be observed during Cd sorption. On the one hand, Cd sorption is slightly enhanced in comparison with $S$ when $A$ content increases under alkaline conditions and high ionic strength ($1 \cdot 10^{-1}$ M). On the other hand, Cd sorption decreases at pH < 4.5 and low ionic strengths ($\leq 1 \cdot 10^{-2}$ M) when $A$ is present in the $A/S$ mixtures. This is due to the dissolution of $A$ and subsequent Al$^{3+}$ competition with Cd$^{2+}$ for cation exchange sites in $S$,

which has a higher effect at low ionic strengths. Thus, the presence of $A$ in the mixtures has two opposite influences on Cd sorption.

The SM in $A/S$ mixtures fit the experimental results by considering the models of pure $A$ and $S$ and adding the cation exchange competition of $Al^{3+}$.

Therefore, the chemistry of suspensions highly influences Cd sorption on $A/S$ mixtures and needs to be analyzed before simulating and evaluating the effectiveness of such mixtures and their potential in remediating a Cd-polluted site.

**Supplementary Materials:** The following supporting information can be downloaded at: https://www.mdpi.com/article/10.3390/min13121534/s1. The supporting information shows: the Cd speciation diagram for the most-used experimental conditions (Figures S1–S3), additional graphs for the Cd sorption model on alumina (Figures S4–S7), and alumina and smectite mixtures (Figure S8).

**Author Contributions:** Conceptualization, N.M., T.M. and U.A.; methodology, N.M. and U.A.; formal analysis, N.M. and U.A.; investigation, N.M., T.M. and U.A.; writing—original draft preparation, N.M.; writing—review and editing, N.M., T.M. and U.A.; supervision, T.M. and U.A.; project administration, N.M., T.M. and U.A; funding acquisition, N.M., T.M. and U.A. All authors have read and agreed to the published version of the manuscript.

**Funding:** This research was partially funded by the NukSiFutur TecRad young investigator group, funded by the German Federal Ministry of Education and research, (ref number 002NUK072), the European Union's Horizon 2020 Research and Innovation Programme under Grant Agreement no. 847593 (EURAD, WP FUTURE), and the Spanish Ministry of Science and Innovation (PID2022–138402NB-C22, ACOMER Project).

**Data Availability Statement:** The data published in this paper is available in the HZDR data repository RODARE using the following https://doi.org/10.14278/rodare.2569.

**Acknowledgments:** The authors acknowledge Trinidad López, Manuel Mingarro, and Jesús Morejón for their support in the laboratory. The authors acknowledge the German Federal Ministry of Education and Research (BMBF), the European Commission, and the Spanish Ministry of Innovation and Science for the project funding.

**Conflicts of Interest:** The authors declare no conflict of interest.

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
