# Peer review of "Cadmium Sorption on Alumina Nanoparticles and Mixtures of Alumina and Smectite: An Experimental and Modelling Study"

_minerals, doi:10.3390/min13121534_

Round 1
Reviewer 1 Report
Comments and Suggestions for Authors
The presented manuscript aims to address a pertinent chemistry-related issue. The paper's main theme has been thoroughly researched and developed by the authors. The research approach that will be used is described. To ensure that the results could be replicated, it would be helpful for the reader if the authors supplied more information on the experimental techniques they utilized. The results in the manuscript are intriguing and add fresh information to the field of inquiry and the authors do a fantastic job connecting their findings to earlier research in the field in the discussion section.
However, in my opinion, in equation 6 on page 3 there is a + sign left, on page 6 the reactions 10 and 11 are not correctly adjusted and in the case of 10, there are problems with the elements involved. Line 234 “The sorption equilibrium is reached after 30 min of interaction”, you could show the graph of the first minutes in which this is demonstrated and not just that of the 7 days.
The most significant results of the study are effectively summarized in the conclusion. It should be emphasized that recent and pertinent work has been referenced. Nevertheless, the writers should have added additional references from the previous two years.
The language employed in the text is appropriate for a scientific article, and it is well written. Before resubmitting, I advise authors to go over and respond to the criticisms made. Once the proposed changes are completed, the manuscript has the potential to make a significant addition to the discipline of chemistry.
Author Response
Attached as pdf file

Reviewer 2 Report
Comments and Suggestions for Authors
Reviewer’s comments on the manuscript: Cadmium Sorption on Alumina Nanoparticles, and Mixtures of Alumina and Smectite: An Experimental and Modelling Study written by Natalia Mayordomo, Tiziana Missana and Ursula Alonso
The reviewed manuscript concerns the analyzing Cd immobilization in g−Al2O3 nanoparticles and in mixtures of FEBEX smectite and g−Al2O3 nanoparticles with a combined experimental and theoretical approach. The manuscript presents very interesting data having high application potential. I really appreciate the aim of study and the way of data presentation. Moreover, results were obtained using variety of analytical techniques. However, the text requires correction because the authors did not avoid mental shortcuts and minor errors. In the reviewer’s opinion minor revision is required.
- page 2, line 54 and further: “In some cases, Cl induces a higher Cd removal” I don't like the fact that when writing about chlorides, the authors use the abbreviation “Cl”, referring to chlorine, and not chlorides. Please, change “Cl” into chloride/ Cl- anions.
- page 2, line 60: change “Clay” into “clay”.
- page 2, line 70: “Cl concentrations” I think you mean the concentration of chloride anions.
- page 2, line 85: “N2 specific surface area (136 m²/g)” It is misunderstanding. You probably mean the specific surface area of g-Al2O3 measured using adsorption-desorption of nitrogen. The same comment line 94.
- page 3, table 1: I am not an English native speaker but some of the words and phrases that the author used are awkward “pH dependency”? It probably should be “pH dependence”. My suggestion is to give this paper to a professional English translator. The same comment table 2.
- page 5: Please add reference after the sentence: “The SM of Cd on A/S mixtures took into account the water chemistry”
- page 8, line 232 and further: What does that unpainted circles at the end of the captions under figures mean?
- page 8, line 235: Dot is missing.
- page 8, 238: Space is missing. The same line 256.
- page 8: How do the authors comment on the weak relationship between Cadmium sorption on alumina nanoparticles at different NaClO4 strengths. The presented discussion is insufficient.
- page 10, line 289, line 303: Editorial mistakes.
- page 11, lines 353/354: editorial mistake. The same line 425.
- page 12: Discussion concerning data presented in Fig. 5 are not convincing. Could you please discuss this topic in more detail. These results can be very useful.
Comments on the Quality of English LanguageMy suggestion is to give this paper to a professional English translator for correction.
Author Response
Attached as pdf file

Reviewer 3 Report
Comments and Suggestions for Authors
I enjoyed reading the manuscript "Cadmium Sorption on Alumina Nanoparticles, and Mixtures of Alumina and Smectite: An Experimental and Modelling Study".
The authors conducted numerous experiments that allowed them to study the influence of various factors on the sorption of Cd by aluminium oxide, smectite and their mixture. The authors created a model to describe the adsorption of Cd from solutions with low concentrations and suggested the participation of ternary anionic complexes of Cd in adsorption at pH < 5.
The work was carried out at a very high scientific level, the data were discussed comprehensively, very competently and deeply analysed. Certainly, the paper is ready for publication.
A few technical remarks.
1.Lines 248-249. The authors write "Cd sorption on alumina increases with increasing pH, being maximum (> 90% Cd sorbed) for pH > 7.4". Figure 2 shows that the maximum sorption is observed in the pH range of 9 -10. The text needs to be edited.
2.Please check the caption of figure S4. Black squares and red circles are not observed in the figure. Figure S4 is missing in the Table of Contents section

Author Response
Attached as pdf file

Reviewer 4 Report
Comments and Suggestions for Authors
The search for adsorbent materials is very necessary to remedy pollution problems. Recently, using nanoparticles of alumina in the sorption process has become more attractive to many scientists because of their unique properties. Alumina nanoparticles are some of the widely studied metal oxides that can remove heavy metal ions from aqueous solutions due to their large surface area, high sorption capacity, and mechanical strength. Hence the manuscript is interesting and meets the purposes of the Journal, and I recommend it for publication. You only need to correct the following sections:
The figure must modify its scale to clearly see the trend of the points
Further study to evaluate the mobility in the subsurface due to the combined effect of its negative charge, stability, and the change in particle sizes, need more explicit.
Comments on the Quality of English LanguageThe search for adsorbent materials is very necessary to remedy pollution problems. Recently, using nanoparticles of alumina in the sorption process has become more attractive to many scientists because of their unique properties. Alumina nanoparticles are some of the widely studied metal oxides that can remove heavy metal ions from aqueous solutions due to their large surface area, high sorption capacity, and mechanical strength. Hence the manuscript is interesting and meets the purposes of the Journal, and I recommend it for publication. You only need to correct the following sections:
Author Response
Please see the attached as pdf file.

Reviewer 5 Report
Comments and Suggestions for Authors
This is a very interesting manuscript, nicely written and well structured. Minor modifications/clarifications, detailed below, are needed prior its publication:
Introduction
Line 70. Add range of Cl concentrations used and indicated the ionic strength buffer used.
Materials and methods
Line 89. Add molecular formula for all listed minerals.
Line 90. Although other reference are mentioned for solid phase characterization information, I would suggest to add in this manuscript a table with FEBEX bentonite detailed composition.
Line 109. Typo in the title “Cadmiumd” instead of “Cadmium”.
Table 4. Instead of using the CHESS 2.4 code definition to report chemical reactions I will suggest using IUPAC standards to define these reactions.
Results and discussions
Figure 1. The model line has no meaning in this figure. The authors are using a surface complexation model which work on a thermodynamic basis, thus it cannot be used to reproduce a kinetic trend. I would suggest removing the line from Figure.
Line 325. The use of an empirical Kd value to fit the experimental results at acid conditions is just a numerical artifact. Authors claim that there aren’t experimental evidences to justify the observed behaviour so I would suggest removing these data from Figure unless they can provide additional proves or justifications of sorption.
Line 375. Are the authors able to provide a quantification of A/S dissolution? Are the authors able to provide a chemical analysis of contact waters to quantify dissolved cations in solution?
Conclusions
Line 428. Authors are mentioning an effect of Ionic Strength because of the cation competition but values used to buffer ionic strength are relatively low. For such argumentation and a clear comparison authors should include sorption data at higher ionic strengths (i.e. 0.5 and 1M).
Supporting information
Cadmium speciation. Authors are encourage to use theoretical calculations to illustrate and justify the effect of Cl on Cd speciation. For example, the use of Pourbaix diagrams (Cl vs pH) or fractional diagrams with variable Cl content will clearly illustrate the effect of Cl on Cd speciation. Same could be performed for carbonate effect on Cd.
Comments on the Quality of English LanguageThe use of language is correct but some inconsistencies should be addressed. A thorough check is recommended.
Author Response
Attached as pdf file

Round 2
Reviewer 4 Report
Comments and Suggestions for Authors
The authors have complied with the requested recommendations and it is ready to be published, it is recommended that they upload the manuscript with the updated version. Regards
Comments on the Quality of English LanguageThe authors have complied with the requested recommendations and it is ready to be published, it is recommended that they upload the manuscript with the updated version. Regards
Author Response
Changes were not required to be done in the second round of revisions by reviewer 4